ten Hacken *et al. Genome Biology*    (2020) 21:266

**SHORT REPORT**    **Open Access**

# High throughput single-cell detection of multiplex CRISPR-edited gene modifications

Elisa ten Hacken[1,2†], Kendell Clement[3,4,5†], Shuqiang Li[3,6], María Hernández-Sánchez[1,7], Robert Redd[8], Shu Wang[9], David Ruff[9], Michaela Gruber[1,10], Kaitlyn Baranowski[1], Jose Jacob[9], James Flynn[9], Keith W. Jones[9], Donna Neuberg[8], Kenneth J. Livak[6], Luca Pinello[3,4,5*] and Catherine J. Wu[1,2,3,11*]

* Correspondence: lpinello@mgh.
harvard.edu; cwu@partners.org
†Elisa ten Hacken and Kendell
Clement contributed equally to this
work.
[3]Broad Institute of MIT and Harvard,
Cambridge, MA, USA
[1]Department of Medical Oncology,
Dana-Farber Cancer Institute,
Boston, MA, USA
Full list of author information is
available at the end of the article

## Abstract

CRISPR-Cas9 gene editing has transformed our ability to rapidly interrogate the functional impact of somatic mutations in human cancers. Droplet-based technology enables the analysis of Cas9-introduced gene edits in thousands of single cells. Using this technology, we analyze Ba/F3 cells engineered to express single or multiplexed loss-of-function mutations recurrent in chronic lymphocytic leukemia. Our approach reliably quantifies mutational co-occurrences, zygosity status, and the occurrence of Cas9 edits at single-cell resolution.

**Keywords:** Genetics, Single cell, Genome editing, Loss-of-function, Mutation, CRISPR-Cas9, chronic lymphocytic leukemia

## Background

Large-scale genetic sequencing studies of human cancers have identified a number of novel putative disease drivers, whose function is beginning to be understood partly as the result of advances in genetic engineering technologies. Among these, CRISPR-Cas9 gene editing has transformed our ability to rapidly interrogate the role of somatic mutations [1, 2], both in vitro in cell lines [3] and in vivo in mouse models [4, 5]. This system is based on the activity of the Cas9 endonuclease that generates double-strand breaks at a target locus with sequence complementarity to a small-guide RNA (sgRNA). Subsequent non-homologous end joining (NHEJ) of the target sequence produces insertion/deletions (indels) that disrupt gene function and thereby recapitulate loss-of-function (LOF) mutations observed in human cancers. In addition to advances in the interrogation of individual cancer drivers, rapidly evolving single-cell sequencing technologies [6–8] can be used to define tumor clonal architecture and mechanisms of clonal evolution of cancer cells, with better sensitivity and resolution than conventional bulk techniques. Despite these advances, tools to quantify the relative abundance of CRISPR-introduced disease drivers at the single-cell level, particularly in the context of multiplexed gene editing, are currently lacking. As most cancers arise and propagate due to the complex interaction of multiple disease drivers, these tools would facilitate

the assessment of the contribution of individual genetic drivers to cellular fitness, their preferential co-occurrences, and the study of clonal evolutionary mechanisms both in cellular and mouse models.

## Results and discussion

Here, we utilize CRISPR-Cas9 gene editing to model common LOF alterations recurrent in chronic lymphocytic leukemia (CLL) [9] (i.e., *TP53*, *ATM*, *CHD2*, *SAMHD1*, *MGA*, *BIRC3*), either individually or in a multiplexed fashion in the murine interleukin 3 (IL-3)-dependent pro-B cell line Ba/F3 [10]. For multiplex gene editing, a qPCR method was developed to detect the number and identity of multiple sgRNAs in single cells. The genome alterations induced by single or multiple sgRNAs were assayed in thousands of individual cells using droplet-based sequestration of single cells, PCR generation of targeted sequencing libraries, and next-generation sequencing (NGS).

As a prelude to multiplex gene editing, control experiments were performed to assess single LOF cell lines. The Ba/F3 cell line was transduced to constitutively express Cas9, then transduced with lentivirus expressing single mCherry-tagged sgRNAs targeting the loci of interest in the murine genome (i.e., *Trp53*, *Atm*, *Chd2*, *Samhd1*, *Mga*, *Birc3*) (Fig. 1a). Editing at the target loci was verified 10 days post-transduction by bulk NGS and CRISPResso2 [11] software analysis, confirming the presence of > 80% gene edits for all 6 targets (Fig. 1b). In order to assess gene editing at the single-cell level and interrogate putative Cas9 off-target activity, single-cell DNA sequencing was performed using the Tapestri system (Mission Bio). A panel of 40 PCR amplicons with an average size of ~ 220 bp was designed to analyze the CRISPR editing target loci in the 6 genes of interest, 22 putative off-target loci (predicted in silico using the CRISPOR algorithm) [12], and 12 internal positive controls. Following multiplex PCR on single cells in barcoded droplets, pooled single-cell amplicon libraries were sequenced, and the adapter and barcode sequences were removed from the sequencing reads during analysis. Reads were assigned to single cells by cell barcode, then aligned to each amplicon using CRISPResso2 [11]. The fraction of modified reads in each cell for each amplicon was calculated and used for downstream analysis (Fig. 1c). An admixture of the six single-edited cell lines was processed using the Tapestri system, generating output with a median of 26 reads aligned per amplicon per cell (Additional file 1: Fig. S1a). Out of 4328 cells analyzed, 3102 (~ 71.7%) had at least 5× coverage for at least 26 gene targets (Additional file 1: Fig. S1b). Positive control amplicons showed a median coverage of 24 reads per amplicon per cell (Additional file 1: Fig. S1c). Analysis of editing in single cells showed modifications in at least one on-target locus in 3357/4328 (77.6%) single cells. Editing was observed at on-target loci in cells with sufficient coverage for *Trp53* ($n = 512/3721$, 13.8%), *Atm* ($n = 619/4289$, 14.4%), *Chd2* ($n = 1051/4312$, 24.4%), *Samhd1* ($n = 590/4023$, 14.7%), *Mga* ($n = 331/3211$, 10.3%), and *Birc3* ($n = 616/4309$, 14.3%) (Fig. 1d), and the off-targets showed substantially lower editing, if any (Additional file 1: Fig. S1d). In order to estimate the presence of doublets/triplets in our output dataset, we counted unique cell barcodes with modifications at more than one on-target editing site (6 on-target sites were considered). Because the input cell population was an admixture of cells edited with only one sgRNA, cell barcodes with editing at more than one on-target site are most likely cell doublets/triplets. Our analysis revealed the presence of 335 cells (7.7%) carrying two gene edits and 13 cells

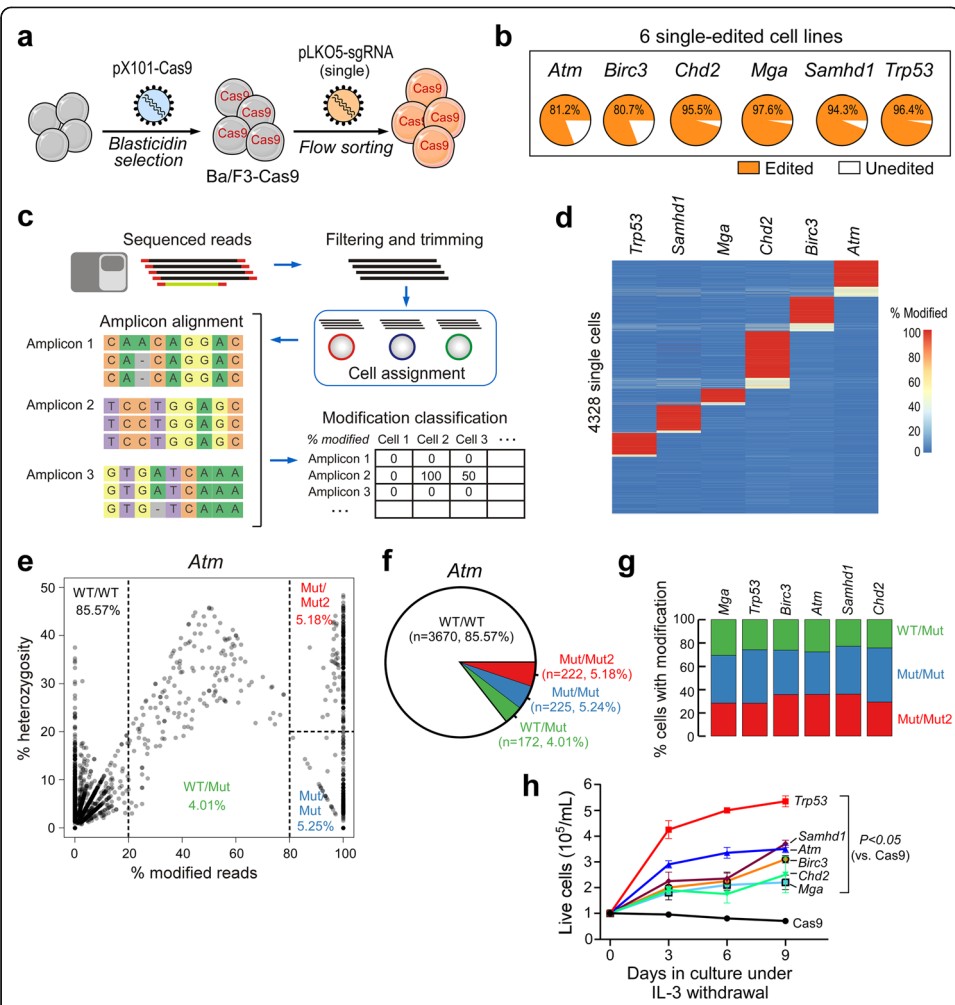

**Fig. 1 a** Schema of generation of single-edited Ba/F3 cell lines. **b** CRISPR-seq NGS validating the presence of gene edits in the 6 independent single-edited cell lines. **c** Single-cell DNA-seq analysis overview. First, sequenced reads are filtered and trimmed, after which they are assigned to individual cells by cell barcodes, and aligned to each amplicon using CRISPResso2. Finally, the modification rates at each editing on- and off-target in each cell are compiled. **d** Heatmap showing modification rates across 4328 cells (rows) at the 6 on-target loci. A scale bar showing the percent of mutated reads ranging from 0% modification (blue color) to 100% modification (red color) is shown. **e** Example of zygosity analysis at one of the on-target loci. Analytical thresholds (described in the "Methods" section) identify WT/WT (purple), WT/Mut (green), Mut/Mut2 (red), and Mut/Mut (blue) cells that are highlighted in the figure. **f** Pie chart showing heterozygous/homozygous subpopulations of gene edits within the *Atm* on-target locus. Numbers and percentage read counts for each sub-group are shown. **g** Cumulative zygosity analyses across the 6 on-targets for cells that have at least one mutated allele (WT/WT are omitted). **h** Absolute counts of live single-edited cell lines (and Cas9 control line) cultured for 9 days under IL-3 withdrawal. *P* value, RM-ANOVA with Dunnett's correction for multiple comparisons

(0.3%) carrying more than two gene edits (Additional file 1: Fig. S1e). We found that multiple edits occurred in cells with both high and low read coverage, so this criterion could not be used to filter cells based on the number of reads (Additional file 1: Fig. S1f).

Zygosity of the edited gene targets was evaluated by calculating the frequency of the second most frequent allele. Cells with < 20% modified alleles were considered WT/WT. Cells with ≥ 20 to < 80% modified alleles were considered heterozygous WT/Mut. For cells with ≥ 80% mutated alleles, those with ≥ 20% second allele were classified as heterozygous Mut/Mut2, while those with < 20% second were classified as homozygous

Mut/Mut (Fig. 1e). At the *Atm* on-target locus (Fig. 1f), upon admixture of the 6 cell lines, the vast majority of cells (85.57%) carry two wild-type alleles, and we could distinguish smaller proportions carrying either one (WT/Mut, 4.01%) or two edited alleles, either in heterozygosity (Mut/Mut2, 5.18%) or in homozygosity (Mut/Mut, 5.25%); a similar percentage distribution of edited sub-groups based on zygosity was observed across all 6 targets (Fig. 1g and Additional file 1: Fig. S1g). Functionally, introduction of single disease drivers conferred survival advantage when cells were cultured in the absence of IL-3 (Fig. 1h), thus confirming the applicability of our tools to the functional evaluation of novel cancer drivers.

In order to generate multiple CRISPR edits in the same cell, cells were transduced with a pool of lentivirus for all six targets to generate a multiplexed Ba/F3 cell line (Fig. 2a). Bulk CRISPR-seq NGS confirmed the presence of variable levels of gene edits for the six gene targets (Fig. 2b). To assess transcription of the sgRNAs of interest in individual Ba/F3 cells, we designed a single-cell qPCR method based on a nested strategy: (1) reverse transcriptase reaction using random primers, (2) pre-amplification using outer primers, and (3) qPCR assays using inner primers run on the Fluidigm Biomark system [13] (Fig. 2c). We analyzed sgRNA expression for 96 multiplexed Ba/F3 cells, detecting the presence of up to 5 sgRNAs/cell. Most cells (44/96) expressed 2 sgRNAs, with different combinatorial assortments (Additional file 1: Fig. S2a-b). To detect the resulting DNA edits, the Ba/F3 multiplexed line was processed on the Tapestri platform to analyze 3429 cells, of which 3200 cells showed gene modifications in at least one of the six on-target loci. Most cells carried either one (1205 cells) or two (1173 cells) gene edits at on-target genes, but we also detected the presence of cells carrying edits in 3 target genes (525 cells) or more (4 gene edits, 138; 5 gene edits, 56; 6 gene edits, 4) (Fig. 2d). When comparing the initial profile (Fig. 2e, left panel) to the one observed upon in vivo passaging of the multiplexed BaF3 line into NSG mice, we observed strong in vivo selection of cells expressing combined *Mga* and *Chd2* mutations, when these were isolated from the splenocytes of NSG recipient mice at euthanasia (Fig. 2e, right panel). The many combinatorial assortments among the 6 targets observed before transplant (Fig. 2f) could no longer be identified after transplant (Fig. 2g), suggesting that cells carrying selected combinatorial traits showed competitive advantage upon in vivo transplantation.

## Conclusions

In summary, these results demonstrate feasibility of single-cell DNA detection of gene edits for common LOFs generated by CRISPR-Cas9 gene editing. In contrast to previously published platforms [6, 8], our method allows assessment of a larger number of cells (> 3000/sample), across multiple Cas9-target loci. Single-cell detection of modifications is particularly important for deciphering the effects of multiplex gene editing, and our results demonstrate that co-transduction with six sgRNAs can generate many combinations of gene modifications. With respect to previously published large-scale functional screens of putative cancer drivers [14, 15], our novel platform uniquely provides the opportunity to study mutational co-occurrences at the single-cell level, thus providing a higher resolution tool for the functional interrogation of disease drivers. We do not envision problems in scaling up this platform to the testing of hundreds or thousands of gene edits, as long as high-titer lentiviral preparations are utilized,

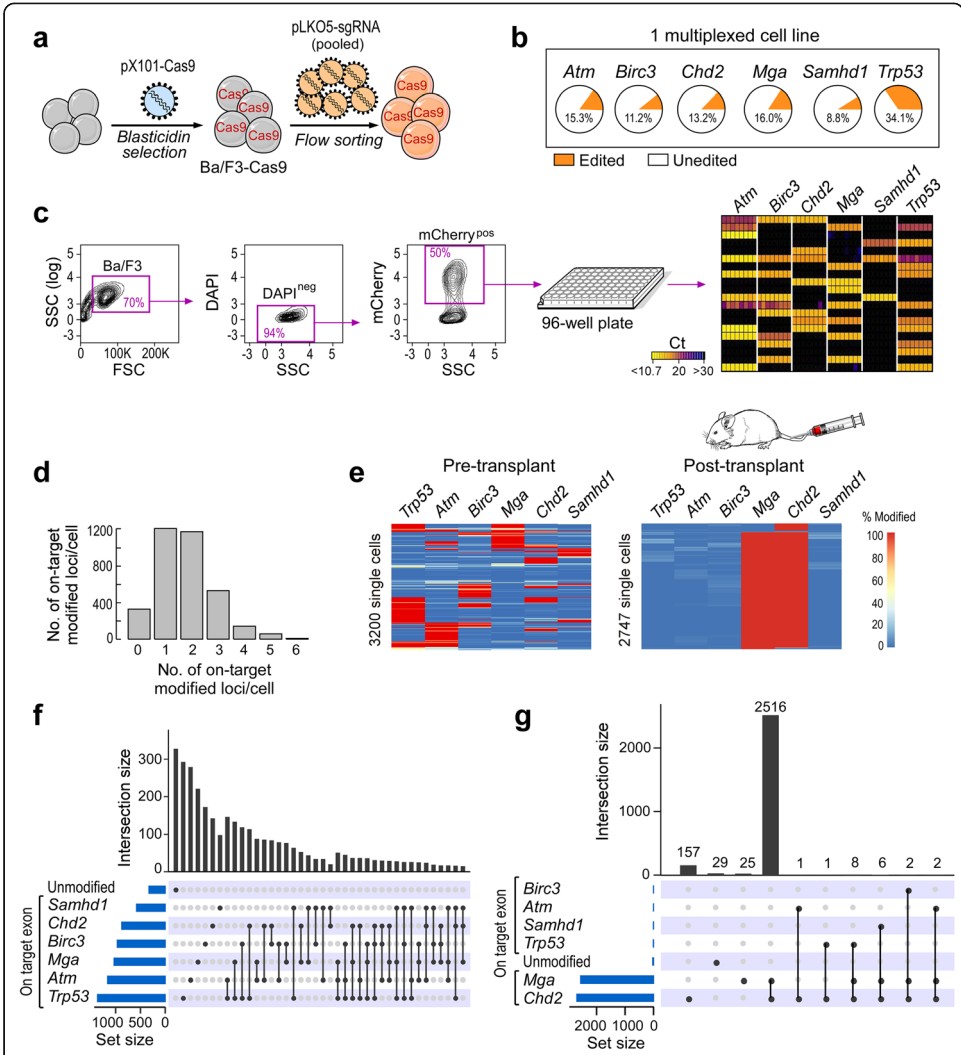

**Fig. 2 a** Schema of generation of a multiplexed Ba/F3 cell line. **b** CRISPR-seq NGS validating the presence of gene edits for all 6 targeted loci. **c** Workflow for single-cell qPCR detection of sgRNAs by Fluidigm Biomark. Single cells from the Ba/F3 multiplexed line were sorted (based on viable, DAPI⁻, and transduced mCherry⁺) into one PCR plate, then analyzed using the Biomark assay. **d** Histogram reporting the number of on-target gene edits detected across 3429 single cells, from the multiplexed-edited Ba/F3 cell line. **e** Modification rates across single cells containing at least one gene edit (rows) for the six on-target loci (columns) using single-cell amplicon sequencing. Cells (n = 3200 pre-transplant, n = 2747 post-transplant) were assayed before and after in vivo passaging, obtained via intravenous injection into NSG mice. Splenocytes were sampled at euthanasia. A scale bar ranging from 0% (blue color) to 100% (red color) modifications is shown. **f** Histogram showing the number of cells containing mutations for the indicated gene combinations. Set size refers to the number of cells for each of the six targets. Intersection size refers to the number of cells for each gene-edit combinatorial assortment. **g** Combinatorial assortments of cells post-transplant

although the required sequence coverage (~ 30× coverage per amplicon per cell) may represent a potential limitation for analysis when utilizing currently available pipelines. Despite these technical advances, we acknowledge that our approach for LOF mutation detection and selection may facilitate stronger disease drivers, as weaker drivers may be selected against both in the process of in vitro culture, or during in vivo passaging. On-going functional examinations will further elucidate how these various combinations of putative cancer drivers affect phenotype. Thus, the analysis of sgRNA co-transduction

and zygosity in single cells reported here supports the use of our multiplexed genome editing system to study the co-dependence and interaction of multiple disease-associated lesions.

## Methods

### sgRNA design and Ba/F3 cell line generation

sgRNAs were designed using the online GFP sgRNA Designer (CRISPRko) design tool (https://portals.broadinstitute.org/gpp/public/analysis-tools/sgrna-design) [16] to target *Atm*, *Birc3*, *Chd2*, *Mga*, and *Samhd1*. sgRNAs were chosen based on the highest editing efficiency and the lowest predicted off-targeting activity. The predicted sgRNA cut sites lie upstream of any protein domain or within the first protein domain according to the UniProt database (https://www.uniprot.org). For *Trp53*, we chose a sgRNA previously reported to have high editing activity in B cells [17]. Sequences of the on-target sgRNAs are detailed in Additional file 2 -Table S1. sgRNAs were cloned into pLKO5.sgRNA.EFS.tRFP (Addgene #57823). Cloning was performed as previously described [18]. Briefly, the pLKO5.sgRNA.EFS.tRFP plasmid containing the coding sequence of mCherry and a cloning site for the sgRNA sequence was digested with Esp3I (BsmBI) (NEB). To clone the sgRNAs into the pLKO5 vector, two complementary oligos for each sgRNA were designed including two 4-nt-overhang sequences; the two complementary oligos were denatured at 95 °C for 5 min, ramp cooled to 25 °C over a period of 45 min to allow annealing, and finally ligated with the linearized pLKO5. Competent cells were transformed with 2 μL of the ligated plasmid, and single colonies were expanded before plasmid extraction. The correct insertion of the sgRNA sequences was confirmed by Sanger sequencing. Lentiviral preparations were then generated via VSV-G and psPAX2 packaging. Ba/F3 cells (500,000/mL in RPMI-10% FBS containing 10 ng/mL of murine IL-3) were transduced with single or pooled lentiviral particles. Three days post-transduction, mCherry$^+$ viable (DAPI$^-$) cells were cell sorted and kept in culture for one additional week, before assessing presence of gene edits by bulk next-generation sequencing (CRISPR-seq NGS). For IL-3 withdrawal studies, single-edited Ba/F3 cells were plated at $10^5$/mL (day 0) in RPMI-10% FBS without murine IL-3 and counted at day 3, day 6, and day 9. *P* values were calculated by repeated-measures ANOVA with Dunnett's correction for multiple comparisons (each group compared to Cas9).

### Bulk CRISPR-seq NGS

To assess presence of gene editing, we designed a multiplexed PCR strategy based on targeted PCR-based amplification of the 6 on-target regions, followed by next-generation sequencing. Briefly, 10 ng of gDNA was amplified by PCR, as follows: 1 cycle of 1 min at 98°C, 35 cycles of 98 °C for 15 s, 65 °C for 30 s, 72 °C for 30 s, and 1 cycle of 5 min at 72 °C. The primers targeted a 200-bp region flanking the sgRNA cut site (Additional file 3 -Table S2). Primers were designed by Primer-BLAST with similar annealing temperatures, thus favoring multiplexed amplification at a similar efficiency. The PCR product was then purified and submitted to next-generation sequencing through the Massachusetts General Hospital (MGH) DNA core. FASTQ files were then

processed by CRISPResso2 [11] software analysis to quantitatively and qualitatively assess presence of gene edits.

### Single-cell qPCR by Fluidigm Biomark

Single mCherry[+] cells were dry sorted into 96-well PCR plates (Eppendorf 0030129512) and stored at − 80 °C. For processing each plate, a Lysis Mix was prepared by combining 952.5 μL DNA Suspension Buffer (10 mM Tris, pH 8.0, 0.1 mM EDTA, Teknova T0221), 2.5 μL 40 units/μL RNaseOUT (Thermo Fisher 10777019), plus 50 μL 10% NP40 (Thermo Fisher PI-28324), and 5 μL Lysis Mix was added to each well. cDNA was synthesized by adding 1 μL Reverse Transcription Master Mix (Fluidigm 100-6299) to each well, incubating at 25 °C for 10 min, 42 °C for 30 min, and 85 °C for 5 min, then placing on ice. A Pre-Amp mix was prepared by combining 208 μL 5× PreAmp Master Mix (Fluidigm 100-5581), 104 μL 500 nM each PreAmp primers, plus 104 μL water, and 4 μL was added to each reaction. The PreAmp primers (Additional file 4 -Table S3) were a mix of 12 forward primers and one reverse primer. The forward primers were specific for the guide RNAs of six mouse genes (*Atm*, *Birc3*, *Chd2*, *Mga*, *Samhd1*, *Trp53*). The reverse primer was specific for the common tracrRNA segment. The conditions for pre-amplification were 95 °C for 5 min, 20 cycles of 96 °C for 5 s, and 60 °C for 6 min, followed by 4 °C hold. An Exonuclease mix was prepared by combining 84 μL Exonuclease I (NEB; M0293L), 42 μL 10× Exonuclease I reaction buffer, plus 294 μL water, and 4 μL was added to each reaction. These reactions were incubated at 37 °C for 30 min, 80 °C for 15 min, then placed on ice. Thirty-six microliters of DNA Suspension Buffer was added to each reaction, and the plate was stored at − 20 °C. The pre-amplified samples were analyzed by qPCR using 96.96 Dynamic Array™ IFCs (Fluidigm BMK-M10-96.96-EG) and the Biomark™ HD System from Fluidigm. Processing of the IFCs and operation of the instruments were performed according to the manufacturer's procedures. A Master Mix was prepared consisting of 420 μL 2× SsoFast™ EvaGreen Supermix with Low ROX (BioRad 172-5211), 42 μL 20× DNA Binding Dye Sample Loading Reagent (Fluidigm 100-5360), plus 18 μL water, and 4 μL of this mix was dispensed to each well of a 96-well assay plate. Three microliters of pre-amplified cDNA sample was added to each well, and the plate was briefly vortexed and centrifuged. Following priming of the IFC in the IFC Controller HX, 5 μL of the cDNA sample + Master Mix was dispensed to each Sample Inlet of the 96.96 IFC. For each of 12 assays, 5 μL 10× Assay mix [5 μM each primer/10 mM Tris-HCl, pH 8.0/1 mM EDTA/0.25% Tween-20 (Thermo Fisher PI-28320)] was dispensed to each Detector Inlet of the 96.96 IFC, with 8 replicates for each assay. The 12 assays consist of a sgRNA-specific forward primer and a common tracrRNA-specific reverse primer, all designed to be nested inside the pre-amplification primers. Primer sequences are in Table S3. After loading the assays and samples into the IFC in the IFC Controller HX, the IFC was transferred to the Biomark HD and PCR was performed using the thermal protocol CRISPR v1.pcl, which consists of a Thermal Mix of 70 °C, 40 min; 60 °C, 30 s; Hot Start at 95 °C, 1 min; PCR cycle of 30 cycles (96 °C, 5 s; 50 °C, 20 s); and melting using a ramp from 60 to 95 °C, at 1 °C/3 s. Data was analyzed using Fluidigm Real-Time PCR Analysis software using the Linear (Derivative) Baseline Correction Method and the Auto (Global) Ct Threshold Method. The Cq values determined were exported to Excel for further processing.

### Tapestri primer design and selection of target loci

A single-cell DNA sequencing panel was designed by the Tapestri primer design algorithm developed for Tapestri multiplex PCR biochemistry and optimized for uniform amplification. A verification and validation process was used to test performance of the primer design algorithm developed on Tapestri chemistry. We designed various small, medium, and large panels. Multiple runs were conducted for each panel with different cell types. We were able to achieve high panel performance and uniformity. For each CRISPR target, a 40-bp window centered on the CRISPR cleavage site was targeted by one amplicon for indel detection. A panel of 40 amplicons with sizes 196–260 bp was designed to target and capture regions of interest in the mouse GRCm38/mm10 genome (6 CRISPR on-target loci, 22 off-targets, 12 mouse exonic positive control loci). With an overall in silico coverage of 99.6%, the panel covers 27 out of 28 CRISPR windows at 100%, and Ints1-off-target window at 72.5%. The genomic coordinates of predicted on-target and off-target regions are reported in Additional file 5 - Table S4. The primers utilized to amplify on- and off-target regions and positive control exons are reported in Additional file 6 - Table S5. Off-target regions were chosen according to the CRISPOR algorithm (http://crispor.tefor.net/crispor.py) [12], based on location within gene coding exonic regions and with a "cutting frequency determination (CFD)" higher than 0.05 used as a criterion for likelihood of cleavage. Five off-targets located in intronic regions of genes relevant for B cell development were included due to the potential functional implications of their gene editing.

### In vivo xenograft studies

Five million Ba/F3-multiplexed cells were resuspended in 100 μL PBS and injected into the tail vein of 8-week-old female NSG mice (NOD.Cg-$Prkdc^{scid}$ $Il2rg^{tm1Wjl}$/SzJ, Jackson Laboratories). Animals were euthanized when evident signs of disease were detected, including body weight loss, hind limb paresis, labored breathing, and splenocytes banked for further analysis. Prior to single-cell DNA sequencing, splenocytes were cell sorted for expression of cell line-derived mCherry$^+$, and cells then analyzed via Mission Bio Tapestri.

### Single-cell mutational analysis

The Tapestri platform generated single-cell DNA libraries that were sequenced using Illumina MiSeq 300 cycle kit. The sequencing parameters used were Read 1 of 150 bp, Read 2 of 150 bp, Index 1 of 8 bp, and Index 2 of 8 bp. We analyzed reads using a custom pipeline to identify genome editing rates at each targeted amplicon in each cell. The pipeline first trims and filters reads. The sequencing libraries contain two 9-nt cell barcodes flanked by two constant regions. The cell barcodes were removed from the read sequence and attached to the read name, and the constant sequences were trimmed. Reads that were too short (less than 46 bp), or with a cell barcode that was too short (less than 18 bp), were discarded. In the next step, cells with sufficient reads were identified. In this study, we only considered cells with 1000 reads, although this minimum cutoff may vary depending on the number of amplicons profiled and the desired read depth (for our 28 amplicons, this translates into roughly 26 reads per amplicon per cell). Reads were assigned to each valid cell and then processed using CRIS

PResso2 [11] in pooled mode with parameters --force_merge -w 2 --min_reads 5. Briefly, a genome index was created using all of the amplicons, and Bowtie2 [19] was used to align reads from a cell to the amplicons. For each amplicon, each read was aligned to the reference sequence. Amplicons with at least 5 reads were considered. Mutations were considered if they overlapped the quantification window which was set to be a 4-bp window centered at the predicted cleavage position (− 3 bp from the PAM [protospacer adjacent motif] sequence of the guide). For each cell, the modification state (either modified or unmodified) of each allele, as well as the frequency of the allele, was recorded. Finally, the number of aligned reads, the percentage of reads modified, and the modification state and frequency of the top two alleles for each cell and each amplicon were summarized in an output table. For each amplicon, each cell was classified into one of four classes: WT/WT, WT/Mut, Mut/Mut, or Mut/Mut2. For each cell, heterozygosity was calculated as the frequency of the second-most-frequent allele. For cells with one dominant allele, the heterozygosity would be near 0%; cells with heterozygous mutations would have a value of around 50%. Cells with < 20% modified alleles were considered WT/WT. Cells with ≥ 20% and < 80% modified alleles were considered heterozygous WT/Mut. For cells with ≥ 80% mutated alleles, those with ≥ 20% heterozygosity were classified as heterozygous Mut/Mut2, while those with < 20% heterozygosity were classified as homozygous Mut/Mut. Doublets or cell barcodes that likely contained DNA from multiple cells were detected in the admixture experiment as cell barcodes with evidence of on-target editing at multiple target sites. As above, we used a threshold of 20% editing for a site to be counted as edited. We note that our doublet rate (7.8%) is in line with the doublet rate reported by Mission Bio (< 8%, https://support.missionbio.com). Previous investigators [7, 20] have reported low allelic dropout (ADO) ranges of ~ 5–10% using the Tapestri system.

## Supplementary information

---

**Additional file 1: Figure S1.** Single cell DNA-seq of singly-edited Ba/F3 cell lines. QC metrics of single-cell DNA-seq analysis by Tapestri including number of sequencing reads and targeted amplicon coverage, analysis of percent modified reads across on and off-targets and zygosity analysis of on-targets. **Figure S2.** Single cell qPCR-based detection of sgRNAs. Analysis of number and co-occurrence of sgRNAs in single Ba/F3 cells, as assessed via qPCR-based detection. (DOCX 689 kb)

**Additional file 2: Table S1.** Sequences of sgRNAs used in this study. (XLSX 35 kb)

**Additional file 3: Table S2.** Primers used for CRISPR-seq bulk NGS. (XLSX 49 kb)

**Additional file 4: Table S3.** Sequences of primers used for sgRNA amplification by Fluidigm Biomark. (XLSX 9 kb)

**Additional file 5: Table S4.** On and off-target genomic loci analyzed in this study. (XLSX 75 kb)

**Additional file 6: Table S5.** Primer sequences and genomic locations of amplified regions by Mission Bio Tapestri. (XLSX 62 kb)

**Additional file 7.** Review history. (PDF 105 kb)

---

### Acknowledgements
The authors thank the Dana-Farber Cancer Institute Flow Cytometry Core and the Massachusetts General Hospital (MGH) DNA core for excellent technical assistance.

### Review history
The review history is available as Additional file 7.

### Peer review information

## Authors' contributions
E.t.H. and K.C. designed the study, performed the experiments, analyzed the data, and wrote the manuscript; S.L. performed the experiments; M.H-S. contributed to the assay design; M.G. and K.B. contributed to the cell line generation; S.W., D.R., J.J., J.F., and K.W.J. provided the reagents and contributed to the study and assay design; D.N. contributed to the study design; K.J.L. contributed to the study and assay design and wrote the manuscript; L.P. and C.J.W. contributed to the study design, co-supervised the project, and wrote the manuscript. The authors read and approved the final manuscript.

## Funding
C.J.W. acknowledges support from the NIH/National Cancer Institute (NIH/NCI) (R01 CA216273, U10 CA180861, P01 CA206978, and P01-CA081534) and is a scholar of the Leukemia and Lymphoma Society. E.t.H. is a Special Fellow of the Leukemia and Lymphoma Society and a Scholar of the American Society of Hematology. S.L. is supported by the NCI Research Specialist Award (R50CA251956-01). L.P. is supported by the National Human Genome Research Institute (NHGRI) Career Development Award (R00HG008399), Genomic Innovator Award (R35HG010717), and CEGS (RM1HG009490). MHS holds a Sara Borrell post-doctoral contract (CD19/00222) from the Instituto de Salud Carlos III (ISCIII) co-founded by Fondo Social Europeo (FSE) "El Fondo Social Europeo invierte en tu futuro". M.G. was supported by a Marie-Curie International Outgoing Fellowship from the European Union (PIOF-2013-624924).

## Availability of data and materials
The datasets used in the current study have been deposited in Sequence Read Archive (SRA) Bioproject accession number PRJNA665752 [21].

## Ethics approval and consent to participate
Not applicable

## Consent for publication
Not applicable

## Competing interests
C.J.W. is equity holder of BionTech, Inc., and receives research funding from Pharmacyclics. S.W., D.R., J.J., J.F., and K.W.J. are employees of Mission Bio, Inc. L.P. has financial interests in Edilytics, Inc. K.C. is an employee, shareholder, and officer of Edilytics, Inc. The interests of L.P. and K.C. were reviewed and are managed by Massachusetts General Hospital and Partners HealthCare in accordance with their conflict of interest policies. All other authors do not have any relevant conflict of interest.

## Author details
[1]Department of Medical Oncology, Dana-Farber Cancer Institute, Boston, MA, USA. [2]Harvard Medical School, Boston, MA, USA. [3]Broad Institute of MIT and Harvard, Cambridge, MA, USA. [4]Molecular Pathology Unit, Center for Cancer Research and Center for Computational and Integrative Biology, Massachusetts General Hospital, Charlestown, MA, USA. [5]Department of Pathology, Harvard Medical School, Boston, MA, USA. [6]Translational Immunogenomics Lab, Dana-Farber Cancer Institute, Boston, MA, USA. [7]IBSAL, IBMCC-Cancer Research Center, University of Salamanca, Salamanca, Spain. [8]Department of Data Science, Dana-Farber Cancer Institute, Boston, MA, USA. [9]Mission Bio, Incorporated, South San Francisco, CA, USA. [10]Department of Internal Medicine I, Division of Hematology and Hemostaseology, Medical University of Vienna, Vienna, Austria. [11]Department of Medicine, Brigham and Women's Hospital, Boston, MA, USA.

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

## 

