## [Review history. (PDF 105 kb) · Genome Biology]

Review History

First round of review

Reviewer 1

Are you able to assess all statistics in the manuscript, including the appropriateness of statistical tests used? There are no statistics in the manuscript.

Comments to author:

In this manuscript, Wu, Pinello and colleagues describe a relatively high-throughput approach to detecting CRISPR gene edits at the single cell level using an approach that combines droplet fluidics, targeted multiplex PCR, and deep-sequencing. They first demonstrate that the approach is effective in quantitating the frequency of single CRISPR gene edits at a targeted locus and then show the approach can also be used for detecting multigene edits. The method is useful and should be easily adopted, as it is based on techniques and technologies already in use. The technique itself is not novel per se, but the work does represent a nice demonstration of how this approach of targeted amplification and sequencing can be used for detecting multi locus CRISPR edits. The study itself is well done, with good controls, and the data is presented well. Though it is a brief report, it would have been useful to provide some additional brief discussion of caveats of the approach. For example, issues of bias in the multiplex PCR biasing results and the challenge of detecting edits in genes that will not have a selective advantage and thus may be in low frequency. These are minor criticisms though. The manuscript does provide an advance to the field and will be of interest to many biologists.

Reviewer 2

Are you able to assess all statistics in the manuscript, including the appropriateness of statistical tests used? No, I do not feel adequately qualified to assess the statistics.

Comments to author:

Tools to quantify the abundance of CRISPR-edited gene modifications, and to quantify zygosity and mutational co-occurrence at the single-cell level are currently lacking. Such an advance would allow investigators to discover which genetic drivers act synergistically to promote cell growth. In this study, Wu and colleagues have developed new tools to generate singly or multiplexed CRISPR-edited cell lines, and the occurrence of Cas9-induced alterations in single cells examined. This methods-based approach identifies mutational co-occurrence, zygosity status, and the occurrence of gene edits at single cell-resolution. This is well done and should be of interest to computational biologists and cancer biologists. However, this study makes no attempt to investigate functional drivers of CLL. These tools may eventually facilitate the assessment of individual genetic drivers to cellular fitness, their preferential co-occurrences, and the study of clonal evolutionary mechanisms in cellular and mouse models. However, this study does not investigate whether the CRISPR-induced mutations affects cellular fitness of Ba/F3 cells or tumors, thus representing a limitation of the study. Several issues need to be addressed before it is suitable for publication.

1. The Ba/F3 system, which has been around for >20 years, has been widely used to validate oncogenes such as BCR-ABL and for characterizing the transforming potential of mutated kinases. The investigators should assess/confirm that LOF mutations in the 6 individual genes selected (TP53, ATM, CHD2, SAMHD1, MGA, and BIRC3) promotes IL-3 independence and/or transformation of Ba/F3 cells to a pool of Ba/F3 cells.

2. Large scale functional screens have previously been performed in Ba/F3 cells, such as Ng et al, Cancer Cell, 2018 and Guo et al, Cancer Research, 2016. The authors should cite these and provide a discussion of how their newly developed tools will identify new functional drivers and how their approach is distinct from these previous studies.
3. In Figure 1D, the analysis of editing in single cells showed modifications in at least one locus in ~82% of cells. Can the authors comment on this efficiency? Doesn't the degree of editing really depend on the lentiviral infection efficiency in a given cell line? Variability is expected between cell lines. Have the authors analyzed editing in single cells in a cell line other than Ba/F3 cells? If so, can the authors provide a discussion on the variability?
4. In Fig 2A, to generate multiple CRISPR edits in the same cell, Ba/F cells were transduced with a pool of lentivirus for all 6 targets. Most cells carried 1 or 2 edits at on-target genes. This suggests a low efficiency of multiplex gene editing. How feasible is their approach if the investigators were to scale up to hundreds or thousands of gene edits?
5. The investigators should provide an application for their approach. They say that functional examination will further elucidate how these various combinations of putative cancer drivers affect phenotype. Do cells with more than 2 gene edits have a growth advantage over cells with 1 or 2 edits?

Authors Response

Referee #1 (Remarks to the Author):

Q1	In this manuscript, Wu, Pinello and colleagues describe a relatively high-throughput approach to detecting CRISPR gene edits at the single cell level using an approach that combines droplet fluidics, targeted multiplex PCR, and deep-sequencing. They first demonstrate that the approach is effective in quantitating the frequency of single CRISPR gene edits at a targeted locus and then show the approach can also be used for detecting multigene edits. The method is useful and should be easily adopted, as it is based on techniques and technologies already in use. The technique itself is not novel per se, but the work does represent a nice demonstration of how this approach of targeted amplification and sequencing can be used for detecting multi locus CRISPR edits. The study itself is well done, with good controls, and the data is presented well. Though it is a brief report, it would have been useful to provide some additional brief discussion of caveats of the approach. For example, issues of bias in the multiplex PCR biasing results and the challenge of detecting edits in genes that will not have a selective advantage and thus may be in low frequency. These are minor criticisms though. The manuscript does provide an advance to the field and will be of interest to many biologists.
Response	We thank the reviewer for the positive feedback about our work. We do agree that our approach for LOF mutation detection may facilitate stronger disease drivers, as weaker drivers may be selected against both in the process of in vitro culture, or during in vivo passaging. We are clarifying this aspect in the discussion (lines 148-150) and we have also included additional details in the Methods section “Tapestri primer design and selection of target loci” (lines 237-242) to clarify criteria used for optimal targeted PCR panel uniformity.

Referee #2 (Remarks to the Author):

Q1	Tools to quantify the abundance of CRISPR-edited gene modifications, and to quantify zygosity and mutational co-occurrence at the single-cell level are currently lacking. Such an advance would allow investigators to discover which genetic drivers act synergistically to promote cell growth. In this study, Wu and colleagues have developed new tools to generate singly or multiplexed CRISPR-edited cell lines, and the occurrence of Cas9-induced alterations in single cells examined. This methods-based approach identifies mutational co-occurrence, zygosity status, and the occurrence of gene edits at single cell-resolution. This is well done and should be of interest to computational biologists and cancer biologists. However, this study makes no attempt to investigate functional drivers of CLL. These tools may eventually facilitate the assessment of individual genetic drivers to cellular fitness, their preferential co-occurrences, and the study of clonal evolutionary mechanisms in cellular and mouse models. However, this study does not investigate whether the CRISPR-induced mutations affects cellular fitness of Ba/F3 cells or tumors, thus representing a limitation of the study. Several issues need to be addressed before it is suitable for publication.
----	--

Response	We thank the reviewer for demonstrating interest in our work, and for providing valuable comments for improvements. We have performed additional assays (see responses to Q2,Q6) and are providing some unpublished observations regarding application of these tools to mouse models (Figures for reviewers only), to address these queries.																																								
Q2	The Ba/F3 system, which has been around for >20 years, has been widely used to validate oncogenes such as BCR-ABL and for characterizing the transforming potential of mutated kinases. The investigators should assess/confirm that LOF mutations in the 6 individual genes selected (TP53, ATM, CHD2, SAMHD1, MGA, and BIRC3) promotes IL-3 independence and/or transformation of Ba/F3 cells to a pool of Ba/F3 cells.																																								
Response	We thank the reviewer for the request. To address this question, we performed IL-3 withdrawal experiments, showing that all introduced lesions can confer IL-3 independent survival, when compared to the untransduced Cas9-only BaF3 cell line. This data is now included as part of Figure 1h, and here below.   <caption>Approximate data from Figure 1h</caption>   Days in culture under IL-3 withdrawal Trp53 Samhd1 Atm Birc3 Chd2 Mga Cas9     0 1.0 1.0 1.0 1.0 1.0 1.0 1.0   3 4.2 2.8 2.2 2.0 1.8 1.8 1.0   6 5.0 3.2 2.5 2.2 2.0 2.0 1.0   9 5.5 3.8 3.2 3.0 2.5 2.2 1.0   	Days in culture under IL-3 withdrawal	Trp53	Samhd1	Atm	Birc3	Chd2	Mga	Cas9	0	1.0	1.0	1.0	1.0	1.0	1.0	1.0	3	4.2	2.8	2.2	2.0	1.8	1.8	1.0	6	5.0	3.2	2.5	2.2	2.0	2.0	1.0	9	5.5	3.8	3.2	3.0	2.5	2.2	1.0
Days in culture under IL-3 withdrawal	Trp53	Samhd1	Atm	Birc3	Chd2	Mga	Cas9																																		
0	1.0	1.0	1.0	1.0	1.0	1.0	1.0																																		
3	4.2	2.8	2.2	2.0	1.8	1.8	1.0																																		
6	5.0	3.2	2.5	2.2	2.0	2.0	1.0																																		
9	5.5	3.8	3.2	3.0	2.5	2.2	1.0																																		
Q3	Large scale functional screens have previously been performed in Ba/F3 cells, such as Ng et al, Cancer Cell, 2018 and Guo et al, Cancer Research, 2016. The authors should cite these and provide a discussion of how their newly developed tools will identify new functional drivers and how their approach is distinct from these previous studies.																																								
Response	We acknowledge previously reported large functional screens carried out in the same cellular system through either ex vivo mutagenesis by Sleeping Beauty transposons (Guo et al., Cancer Res 2016) or lentiviral-based platforms (Ng et al Cancer Cell 2018) with functional read-outs including ability to xenograft NSG mice and in vitro survival upon IL3 withdrawal. Despite these studies were carried in a larger scale than ours, they fail to provide a system in which the functional interplay of disease drivers can be interrogated in individual cells. Through our platform, we are able not only to model multiplexed alterations, but also to interrogate their co-occurrence at the single-cell level. Our novel tools can be applied further to functional in vivo or in vitro analyses of cancer drivers, as we are currently assessing in novel mouse lines. We have discussed this point in the revised text (lines 144-147), and added new data as part of Figure 2e-left panel and 2g (and shown here below), showing in vivo selection of cells carrying combined Mga and Chd2 mutations, when harvested from the spleen of NSG mice transplanted with the multiplexed edited BaF3 cell line.																																								

	2e-left  g Intersection size 2000 1000 0 157 29 25 2516 1 1 8 6 2 2 On target exon Birc3 Atm Samhd1 Trp53 Unmodified Mga Chd2 2000 1000 0 Set size
Q4	In Figure 1D, the analysis of editing in single cells showed modifications in at least one locus in ~82% of cells. Can the authors comment on this efficiency? Doesn't the degree of editing really depend on the lentiviral infection efficiency in a given cell line? Variability is expected between cell lines. Have the authors analyzed editing in single cells in a cell line other than Ba/F3 cells? If so, can the authors provide a discussion on the variability?
Response	We thank the reviewer for this important technical concern. We would like to clarify that we have generated these lines upon transduction with high-titer lentivirus, followed by a 3-day in vitro culture and cell sorting of mCherry⁺ (sgRNA-expressing cells), which are then further kept in culture for at least a week before assessment of editing efficiency by targeted NGS (see Methods section). We therefore exclude that the editing efficiency is dependent on the lentiviral transduction per se, but rather on the different efficiency of the utilized sgRNAs, with Atm and Birc3 showing a slightly lower activity when compared to the rest.
Q5	In Fig 2A, to generate multiple CRISPR edits in the same cell, Ba/F cells were transduced with a pool of lentivirus for all 6 targets. Most cells carried 1 or 2 edits at on-target genes. This suggests a low efficiency of multiplex gene editing. How feasible is their approach if the investigators were to scale up to hundreds or thousands of gene edits?
Response	response removed due to presence of unpublished data.
Q6	The investigators should provide an application for their approach. They say that functional examination will further elucidate how these various combinations of putative cancer drivers affect phenotype. Do cells with more than 2 gene edits have a growth advantage over cells with 1 or 2 edits?

Response	We thank the reviewer for this important remark. We are now adding new data as part of Figure 2e and 2g , showing in vivo selection of cells carrying combined Mga and Chd2 mutations when the BaF3-multiplexed line was transplanted in vivo into NSG mice.
----------	---

Second round of review

Reviewer 2

The authors have sufficiently addressed my concerns, although it would be nice if they added a discussion of the following to the revised manuscript (in response to my Q5):

"We do not envision problems in scaling up this platform to the testing of hundreds or thousands of gene edits, as long as high-titer lentiviral preparations are utilized, although the required sequence coverage (~30x coverage per amplicon per cell) may represent a potential limitation for analysis when utilizing currently available pipelines."